# Enhanced production of $^{60}$Fe in massive stars

A. Spyrou [1,2] ✉, D. Richman[1,2], A. Couture [3], C. E. Fields[3,4], S. N. Liddick[1,5], K. Childers[1,5], B. P. Crider [1,6], P. A. DeYoung [7], A. C. Dombos [1,2], P. Gastis[3,8], M. Guttormsen [9], K. Hermansen[1,2], A. C. Larsen [9], R. Lewis[1,5], S. Lyons [1,10], J. E. Midtbø[9], S. Mosby[3], D. Muecher[11], F. Naqvi[1,12], A. Palmisano-Kyle[1,2,13], G. Perdikakis[8], C. Prokop[1,3], H. Schatz[1,2], M. K. Smith[1], C. Sumithrarachchi[1] & A. Sweet[14]

Massive stars are a major source of chemical elements in the cosmos, ejecting freshly produced nuclei through winds and core-collapse supernova explosions into the interstellar medium. Among the material ejected, long-lived radioisotopes, such as $^{60}$Fe (iron) and $^{26}$Al (aluminum), offer unique signs of active nucleosynthesis in our galaxy. There is a long-standing discrepancy between the observed $^{60}$Fe/$^{26}$Al ratio by γ-ray telescopes and predictions from supernova models. This discrepancy has been attributed to uncertainties in the nuclear reaction networks producing $^{60}$Fe, and one reaction in particular, the neutron-capture on $^{59}$Fe. Here we present experimental results that provide a strong constraint on this reaction. We use these results to show that the production of $^{60}$Fe in massive stars is higher than previously thought, further increasing the discrepancy between observed and predicted $^{60}$Fe/$^{26}$Al ratios. The persisting discrepancy can therefore not be attributed to nuclear uncertainties, and points to issues in massive-star models.

Unique signatures of supernova explosions can come from long-lived radioisotopes (with half-lives of the order of a million years). These freshly synthesized isotopes are ejected by the supernova and live long enough to either travel all the way to the solar system or emit radiation that does. Two such radioisotopes are $^{60}$Fe and $^{26}$Al with half-lives on the order of a million years. They have been detected in a number of Earth and space-based measurements: presolar stardust grains[1], cosmic rays[2,3], γ-ray measurements of radioactivities in the galaxy[4–10], material deposited in deep-ocean crusts[11–18] and on the surface of the moon[19], indirect signatures in meteorites of their presence during the formation of the solar system[20,21], and more. In addition to individual detections, the $^{60}$Fe/$^{26}$Al ratio has been identified as an excellent probe to study nucleosynthesis in massive stars[10,22]. This ratio, extracted from γ-ray observations, can constrain stellar mixing processes and rotation

associated with slow neutron capture process nucleosynthesis, as well as explodability of supernova models[10]. The latter impacts predictions of the neutron star and black hole mass distributions probed by gravitational waves. The ratio has also been used as a constraint in the study of the origin of the solar system, its possible pollution by a nearby supernova[10], and its place and evolution within the local super bubble[23]. While accurate measurements of this ratio exist in the Galaxy and the early solar system[9], models tend to strongly overestimate it[22,24] and this discrepancy has been an unresolved mystery for several decades.

Both isotopes are predominantly produced in massive stars[10,22,24,25], with small contributions from other sites such as novae, asymptotic giant branch (AGB) stars, type Ia supernovae, and electron capture supernova[26–28]. However, the two isotopes are synthesized in

[1]Facility for Rare Isotope Beams, Michigan State University, East Lansing, MI, USA. [2]Department of Physics and Astronomy, Michigan State University, East Lansing, MI, USA. [3]Los Alamos National Laboratory, Los Alamos, NM, USA. [4]Steward Observatory, University of Arizona, Tucson, AZ, USA. [5]Department of Chemistry, Michigan State University, East Lansing, MI, USA. [6]Department of Physics and Astronomy, Mississippi State University, Mississippi, MI, USA. [7]Department of Physics, Hope College, Holland, MI, USA. [8]Department of Physics, Central Michigan University, Mount Pleasant, MI, USA. [9]Department of Physics, University of Oslo, Oslo, Norway. [10]Pacific Northwest National Laboratory, Richland, WA, USA. [11]Institut für Kernphysik der Universität zu Köln, Köln, Germany. [12]Department of Nuclear Engineering, Texas A&M University, College Station, TX, USA. [13]Physics Department, University of Tennessee, Knoxville, TN, USA. [14]Lawrence Livermore National Laboratory, Livermore, CA, USA. ✉e-mail: spyrou@frib.msu.edu

different parts of the massive star. While both are produced in the inner layers of the star, which are only ejected during the supernova explosion, [26]Al is also produced in H-burning in the outer layers of the star. H-burning material can be ejected earlier due to stellar winds, which adds to the [26]Al yield. For these reasons, the [60]Fe/[26]Al ratio is a powerful tool for understanding the evolution and explosion of massive stars: if both isotopes are produced by the same source(s), then the source distance, location, and number cancel out, giving direct access to the stellar yield ratio right after the supernova explosion[10]. The same arguments could be applied to other ratios of supernova products, however, what makes the [60]Fe/[26]Al ratio unique is the fact that it involves isotopes and not elements. Ratios of elements would have contributions from multiple stellar processes, while the [60]Fe/[26]Al ratio provides a direct connection to the evolution of the massive star, stellar winds, and supernova explosion mechanisms. In fact, because the contributions of the two isotopes come from different parts of the star, a robust model that can match the [60]Fe/[26]Al ratio indicates a good description of the stellar environment across a wide range of stellar material, which would be a significant accomplishment for the field.

The importance of the [60]Fe/[26]Al ratio has been discussed extensively in the literature. For example, parameters such as stellar rotation and explodability (the ability of the star to undergo explosion) have been shown to impact the final [60]Fe/[26]Al ratio values[10,29]. In these studies, a common theme appears in the discussion of model uncertainties, namely the uncertain nuclear reaction networks that produce/destroy the two relevant isotopes[10,22,24,30,31]. The nuclear reaction uncertainties related to the synthesis of [26]Al are less extensive and were discussed in detail by Diehl et al.[10].

The nuclear reaction network that produces [60]Fe is relatively small (Fig. 1(a) inset). It consists of a series of neutron-capture reactions starting at the stable isotope [58]Fe, which compete with either β decays or (p,n) reactions, depending on the astrophysical conditions[31]. In addition, the [22]Ne(α,n)[25]Mg reaction is the dominant source of neutrons in massive stars[32]. Most reactions are well constrained as discussed in the recent review by Diehl et al.[10]. The most impactful and simultaneously most uncertain reaction in this network is the [59]Fe(n,γ)[60]Fe reaction. This reaction is the dominant [60]Fe production mechanism, and the [60]Fe yield was shown to scale linearly with the reaction cross section[30]. It is challenging to measure this neutron-capture directly in the lab due to the short half-life of the target nucleus [59]Fe (44 days)[33]. Therefore, indirect techniques have been used in the past to provide experimental constraints[34,35]. Each of the previously used techniques has its own limitations and uncertainties, however, they both have a common blind spot, namely the low-energy behavior of the γ-ray strength function (gSF).

The gSF represents the reduced probability of the nucleus to emit a γ ray of certain energy and multipolarity[36]. It is one of the most essential quantities used in calculating neutron-capture reaction cross-sections of heavy nuclei[37]. The gSF has been studied for many decades, both experimentally and theoretically, mostly for stable isotopes. During the last decade, measurements of the gSF at low energies revealed a new phenomenon[38], the so-called "low-energy enhancement" or "upbend". The low-energy enhancement was shown to have a dipole character[39], but it is still unclear whether it is of electric (E1) or magnetic (M1) nature[40,41]. Theoretical and experimental investigations show a dependence of the low-energy enhancement on the underlying nuclear structure[41,42], becoming more significant near closed nuclear shells and gradually being reduced in more deformed nuclei. The impact of the low-energy enhancement on neutron-capture reactions was also investigated[43], however its effects vary strongly from reaction to reaction.

Here we present an experimental investigation of the gSF in [60]Fe and its impact on the [59]Fe(n,γ)[60]Fe reaction cross-section. With our measurement we show that the reaction cross section is significantly higher than previously thought, which leads to the conclusion of an enhanced [60]Fe production in massive stars. Our result shows that the discrepancy between models and observations in the [60]Fe/[26]Al ratio persists despite the stronger nuclear physics constraints.

## Results

We performed an experiment to investigate the gSF in [60]Fe, especially at low energies using the β-Oslo method[44,45] (see Methods section). The resulting gSF is shown in Fig. 1b (red squares) together with the extrapolation of the previous [60]Fe study[34] (solid black line) and two measurements of the [56]Fe isotope for comparison[38,39]. It can be seen that similar to [56]Fe, our results show the presence of a significant low-energy enhancement. This, in turn, results in a significant increase of the [59]Fe(n,γ)[60]Fe reaction Maxwellian Averaged Cross Section (MACS)

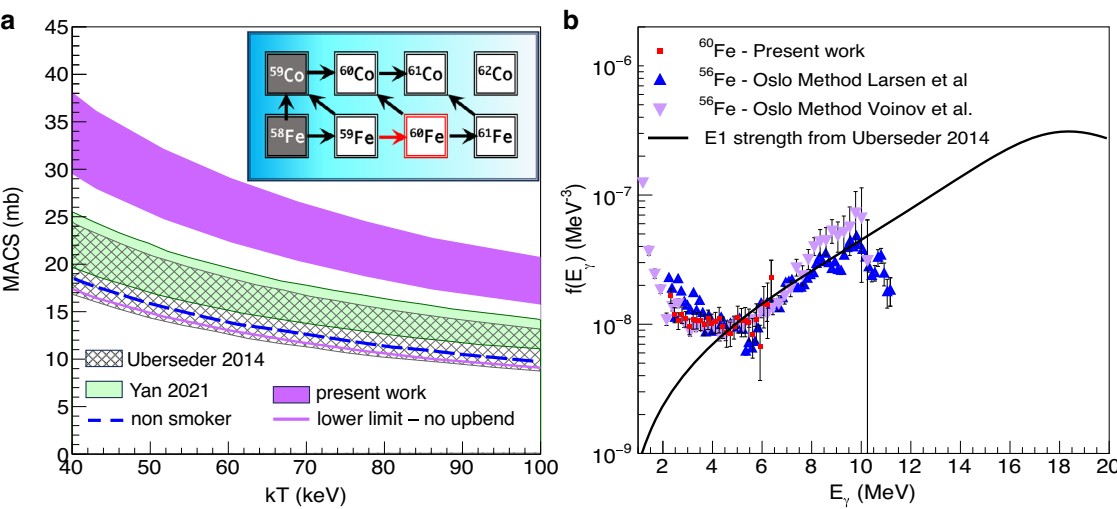

**Fig. 1 | Experimental results. a** Maxwellian Averaged Cross Section (MACS) of the [59]Fe(n,γ)[60]Fe reaction as a function of neutron energy. The hatched and light-green bands represent previous results from Uberseder et al.[34] and Yan et al.[35]. The default MACS used in astrophysical calculations from the non-smoker reaction code is shown as a dashed blue line. The results of the present work are represented by the purple band. For comparison purposes, the lower limit of our results without including the low-energy enhancement in the gSF is shown as a solid purple line. **a** inset: reaction network for the production/destruction of [60]Fe in a massive star. **b** γ-ray strength function showing the presence of a low-energy enhancement in [60]Fe, compared to [56]Fe from previous works[38,39]. The vertical lines crossing each data point represent the uncertainties of the measurements.

(Fig. 1a) compared to the previous measurements[34,35] and compared to the recommended theoretical calculations (non-smoker)[46] used in astrophysical models. More specifically, our lower and upper limits are factors of 1.6 and 2.1 higher than the recommended value. As a consistency check, we performed the MACS calculation removing the low-energy enhancement from our gSF (Fig. 1a, purple solid line) and keeping all other parameters the same. This calculation enables a more comparison with the Uberseder et al. measurement. Although other parameters, like the NLD, are not identical, it can be seen in Fig. 1a that by removing the low-energy enhancement the two measurements are consistent. The main difference between our result and the one from Uberseder et al. comes from the fact that the latter measurement was not sensitive to the presence of low-energy enhancement. The study by Yan et al.[35] also measured at higher energies and used theoretical models to extrapolate to the astrophysically relevant energy. In their gSF extrapolation they also assumed no low-energy enhancement, which could explain the lower cross-section compared to the present work.

## Discussion

To investigate the impact of the new $^{59}$Fe(n,γ)$^{60}$Fe reaction cross section on the production of $^{60}$Fe we ran astrophysical calculations using the lower and upper limits of the new rate (Fig. 2). We evolved solar metallicity stellar models with initial zero-age main-sequence mass of 15, 20, and 25$M_\odot$ using the one-dimensional stellar evolution toolkit, Modules for Experiments in Stellar Astrophysics (MESA)[47,48]. For comparison, we also evolved a baseline model using the default choice of the $^{59}$Fe(n,γ)$^{60}$Fe reaction rate in MESA adopted from Rauscher & Thieleman[46] using the non-smoker Hauser-Feshbach model. Figure 2 shows the mass fraction profiles from the resulting 15$M_\odot$ and 20$M_\odot$ MESA stellar models as a function of Lagrangian mass coordinates for $^{60}$Fe at the pre-supernova stage (at the start of iron core-collapse). In agreement with Jones et al.[30] we find that the dominant pre-supernova regions for $^{60}$Fe production is primarily in the He-, C-, and Ne-burning shell regions. The results from the 25$M_\odot$ models are very similar.

As shown in Fig. 2, in agreement with previous studies[30], the production of $^{60}$Fe both during the massive star evolution and during the supernova depends strongly on the $^{59}$Fe(n,γ)$^{60}$Fe reaction rate. In addition, based on the study by Jones et al.[30], the increase in the $^{59}$Fe(n,γ)$^{60}$Fe reaction rate is expected to propagate throughout the stellar evolution and supernova explosion and impact the final ejected $^{60}$Fe yield. Such a significant increase in the production and ejection of $^{60}$Fe in massive stars has crucial implications in the interpretation of the different observations of $^{60}$Fe mentioned earlier, such as in ocean sediments, the surface of the moon, stardust, and cosmic rays. Here we focus on the impact on the $^{60}$Fe/$^{26}$Al ratio.

The presently accepted value of the $^{60}$Fe/$^{26}$Al ratio within the galaxy, based on γ-ray observations[9], is 0.184+/− 0.042. Astrophysical calculations predict a wide range of values for this ratio. Most massive star models[10,22,24] tend to overpredict this ratio by approximately a factor of 3-10. The disagreement between models and observations is often attributed to uncertain nuclear physics input in the models[9,22]. A recent investigation of the $^{59}$Fe β-decay rate in stellar environments indicated a reduction in the $^{60}$Fe/$^{26}$Al ratio, bringing the models closer to the observations[31]. However, our results point to an increased production of $^{60}$Fe, resulting in a higher value of the $^{60}$Fe/$^{26}$Al ratio in models. The firmer constraints on the nuclear reaction network from the present work show that a nuclear solution to this problem is unlikely. This suggests that the solution to the puzzle should come from the description of the astrophysical environment and processes that affect the two isotopes.

The importance of $^{60}$Fe and $^{26}$Al can further be highlighted when considering that they are produced in different parts of the massive star. The yield of $^{26}$Al in stellar winds was shown to be affected by the mass of the star, the stellar rotation, and the presence of a companion[10,49]. On the other hand, since $^{60}$Fe is ejected only by the supernova, its yield is intricately connected to the explodability of the star, which in turn is connected to black hole formation mass distribution. If a massive star does not explode, then the contribution to $^{60}$Fe is practically zero. Supernova studies exhibit large variation in stellar explodability, which results in significant variation of the $^{60}$Fe/$^{26}$Al ratio[24,29]. For example, Pleintinger et al.[29] found that the $^{60}$Fe/$^{26}$Al ratio is significantly reduced in the case where no supernova exploded, thus no material ejection, above 2 5$M_\odot$[50]. Limongi and Chieffi[24] also showed that for some of their models, the $^{60}$Fe/$^{26}$Al ratio decreased for lower mass limits. Therefore, the increased production of $^{60}$Fe found in the present work can be balanced by assuming a lower mass limit for supernova explosions in the models.

A second stellar parameter found to affect the $^{60}$Fe/$^{26}$Al ratio significantly is stellar rotation. Increased stellar rotation was found to increase $^{26}$Al yields due to increased mass loss[50]. However, stellar rotation was shown to have an even larger impact on $^{60}$Fe yields. This is because stellar rotation causes an enlarged C-burning shell, which in turn increases neutron production[10]. Since neutron-capture reactions are the dominant mechanism of producing $^{60}$Fe, stellar rotation causes an increase in $^{60}$Fe yield, larger than the increase found for $^{26}$Al. As a result, the calculated $^{60}$Fe/$^{26}$Al ratio when including stellar rotation is higher than for non-rotating stars[29]. Therefore, the increased $^{60}$Fe production found in the present work, which causes an even higher $^{60}$Fe/$^{26}$Al ratio than previously predicted, cannot be explained by introducing stellar rotation to the models.

Additional stellar parameters can affect the $^{60}$Fe/$^{26}$Al ratio and further investigations are needed to shed light on this puzzle. One example that was shown to have a significant impact is the explosion energy[40]. Jones et al.[40] showed that the dependence of the $^{60}$Fe/$^{26}$Al is different for 15, 20, or 25$M_\odot$ stars, therefore additional studies are needed to understand the impact in the overall $^{60}$Fe/$^{26}$Al ratio.

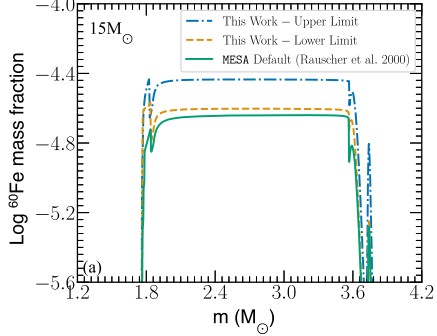
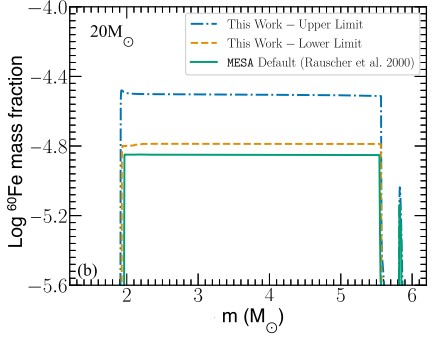

**Fig. 2 | $^{60}$Fe production in massive star simulations. a** Calculation of a 15$M_\odot$ star, and **b** of a 20$M_\odot$ star. The green solid line represents calculations done using the default MACS for the $^{59}$Fe(n,γ)$^{60}$Fe reaction[46], while the blue dot-dashed and yellow dashed lines show the results using the upper and lower limits from the present work.

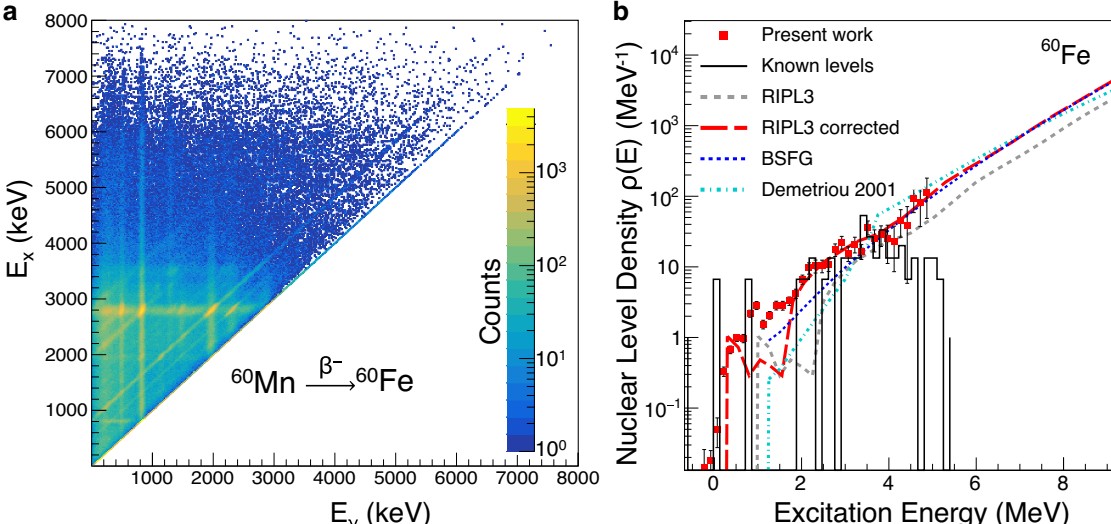

**Fig. 3 | Raw experimental data and NLD for $^{60}$Fe. a** Two dimensional raw matrix showing the γ-ray detection from the β decay of $^{60}$Mn into $^{60}$Fe. **b** NLD of $^{60}$Fe extracted in the present work (red squares) compared to the known discrete levels and theoretical calculations. Vertical lines crossing each of the red squares represent the uncertainties of the present measurement. RIPL3 (grey dashed line) refers to the recommended NLD from the RIPL3 library[63]. RIPL3 corrected (dashed red line) corresponds to the recommended NLD, adjusted to match the experimental data. The blue dotted line corresponds to the NLD calculated using the Back Shifted Fermi Gas model (BSFG)[60], while the cyan dot-dashed line shows the NLD calculated by Demetriou et al.[61].

In the present work, we investigated the production of $^{60}$Fe in massive stars. Our results provide the most complete estimate of the $^{59}$Fe(n,γ)$^{60}$Fe reaction, including the low-energy enhancement in the γ-ray strength function. We found a higher reaction rate compared to previous measurements and the recommended theoretical value used in astrophysical models. The increase in the reaction rate resulted in an increase in the production of $^{60}$Fe in massive stars by almost a factor of 2. We investigated the impact of our results on the $^{60}$Fe/$^{26}$Al ratio, which was identified as a sensitive probe for exploring stellar evolution and supernova dynamics. Before our measurement, the discrepancy between observations and theoretical models in the value of the $^{60}$Fe/$^{26}$Al ratio was attributed to uncertain nuclear physics. While uncertainties in the nuclear physics aspects still remain, our result removes one of the most significant uncertainties in the $^{60}$Fe production. However, the discrepancy persists and is even larger. The solution to the puzzle must come from stellar modeling by, for example, reducing stellar rotation, assuming smaller explodability mass limits for massive stars, or modifying other stellar parameters.

## Methods

The goal of the experiment was to constrain the $^{59}$Fe(n,γ)$^{60}$Fe reaction. In the absence of a direct measurement, experimental techniques focus on constraining the nuclear properties used by theory to calculate the reaction cross section using the Hauser-Feshbach statistical model[51]. These properties are the neutron-nucleus optical model potential (a description of the interaction between the neutron and the nucleus), the nuclear level density - NLD (the number of energy levels per unit energy as a function of excitation energy, spin, and parity), and the γ-ray strength function - gSF (the probability to emit a γ ray of a particular energy and multipolarity). For a recent review of indirect neutron capture constraints for astrophysical processes the reader is referred to Larsen et al.[37]. For neutron capture reactions near the valley of stability the optical model potential is relatively well constrained[52,53]. The other two quantities, however, namely the NLD and gSF are significantly more uncertain, reaching up to two orders of magnitude variation in their theoretical prediction[45]. Here we focus on constraining the NLD and gSF for $^{60}$Fe and in this way constrain the $^{59}$Fe(n,γ)$^{60}$Fe reaction cross section.

In the present work we used a $^{60}$Mn radioactive beam to populate excited states in $^{60}$Fe via β decay and extract its NLD and gSF. To do this we applied the β-Oslo method[44,45], a variation of the traditional Oslo method[54–56], which extracts the NLD and gSF in nuclei populated using various nuclear reactions. The β-Oslo method populates the nucleus of interest using β decay, which was introduced ten years ago for constraining neutron-capture reactions on nuclei far from stability. Here, we use the β-Oslo method to extract the NLD and gSF of $^{60}$Fe and constrain the cross section of the $^{59}$Fe(n,γ)$^{60}$Fe reaction.

The experiment took place at the National Superconducting Cyclotron Laboratory at Michigan State University. A $^{64}$Ni primary beam was accelerated through the Coupled Cyclotron Facility to an energy of 140 MeV/u and impinged on a 510 mg/cm$^2$ thick Be target. The produced cocktail beam centered around $^{60}$Mn was separated using the A1900 fragment separator[57] and delivered to the gas stopping facility[58], where it was slowed down and purified. A pure $^{60}$Mn beam at an energy of 30 keV was implanted into a Si surface barrier detector. The $^{60}$Mn beam consisted of 59(1)% ground state ($J^\pi = 1^+$, $T_{1/2} = 0.28$ s) and 41(1)% isomeric state ($J^\pi = 4^+$, $T_{1/2} = 1.77$ s) (https://www.nndc.bnl.gov). The identification of the two different states was done based on unique γ-ray signatures and their different half-lives. Both states β decay into different excited states of $^{60}$Fe, and the $1^+$ state also decays directly to its ground state.

The β-decay electrons were detected in the Si detector, and the emitted γ rays were detected in the Summing NaI(Tl) (SuN) detector[59]. SuN is a cylindrical total absorption spectrometer segmented into eight optically isolated segments. The total energy deposited in SuN (recorded as the event-by-event sum of all segments) provides the excitation energy, $E_x$, of $^{60}$Fe populated in each β decay. The individual segments are sensitive to the energy of the individual γ rays, $E_\gamma$, emitted during the deexcitation of each $E_x$. For our analysis we use a two-dimensional (2D) matrix of $E_x$ vs $E_\gamma$, which includes all γ rays detected in our experiment following the β decay of $^{60}$Mn into $^{60}$Fe (Fig. 3). The matrix in Fig. 3a shows strong population of discrete states up to $E_x = 3$ MeV. This region was not included in our analysis to avoid strong transitions between discrete levels as the method is not valid for non-statistical γ decay.

The starting point for the β-Oslo method analysis was the 2D γ-ray matrix mentioned earlier ($E_x$ vs $E_\gamma$). The matrix was first unfolded

**Table 1 | MACS upper and lower limits at relevant temperatures**

| kT (keV) | 80 | 85 | 90 | 95 | 100 |
|---|---|---|---|---|---|
| Upper limit (mb) | 24.0 | 23.1 | 22.2 | 21.4 | 20.8 |
| Lower limit (mb) | 18.3 | 17.5 | 16.9 | 16.3 | 15.8 |

**Table 2 | Reaction rate upper and lower limits as a function of temperature**

| T (GK) | Reaction Rate (cm³s⁻¹mol⁻¹) | |
|---|---|---|
| | Upper Limit | Lower Limit |
| 0.1 | 8.2E + 06 | 6.5E + 06 |
| 0.15 | 7.5E + 06 | 5.9E + 06 |
| 0.2 | 7.1E + 06 | 5.6E + 06 |
| 0.3 | 6.7E + 06 | 5.3E + 06 |
| 0.4 | 6.5E + 06 | 5.0E + 06 |
| 0.5 | 6.3E + 06 | 4.9E + 06 |
| 0.6 | 6.1E + 06 | 4.7E + 06 |
| 0.7 | 6.0E + 06 | 4.6E + 06 |
| 0.8 | 5.9E + 06 | 4.5E + 06 |
| 0.9 | 5.7E + 06 | 4.4E + 06 |
| 1 | 5.6E + 06 | 4.3E + 06 |

with the response of the SuN detector[55], and then, following an iterative subtraction process[54], the primary γ-ray distribution was extracted. Primary γ rays are the first γ rays emitted in the deexcitation of an $E_x$ bin, and their distribution depends on the functional form of the NLD and gSF[56]. The final step in this analysis was the normalization of the NLD and gSF using external information in order to extract absolute values for these quantities. Here we used the broad range of known discrete levels, up to ≈4.0 MeV (Fig. 3) to normalize our NLD, which allowed us to fix the slope both for the NLD and the gSF. Because the $^{60}$Mn beam consisted of both the $1^+$ ground state and the $4^+$ isomeric state, the populated NLD corresponded to a broad spin distribution. Assuming allowed β decays from the two initial spins, followed by dipole γ-ray emission, we expect to populate states in $^{60}$Fe with spins 0 – 6 of both parities. This range overlaps completely with the spin range expected to be populated in the neutron-capture on the $3/2^-$ ground state of $^{59}$Fe. It should be noted that the $1^+$ ground state of $^{60}$Mn feeds the $^{60}$Fe ground state with a significant probability. High energy electrons from this gs-gs decay create a background in the SuN detector and for this reason, the primary γ-ray component that feeds the $^{60}$Fe ground state was excluded from the analysis.

The resulting NLD is shown in Fig. 3b (red squares) together with the known discrete levels (black solid line). For comparison, three commonly used theoretical NLD models are shown as well. The Back Shifted Fermi Gas (BSFG) is a phenomenological model often used in global NLD calculations[60]. The BSFG model seems to be in reasonable agreement with the experimental results in the statistical region. Other commonly used models are two semi-microscopic models[61,62], which are normalized to experimental NLD data where available, or to known discrete levels. In Fig. 3b the model labeled "Demetriou 2001"[61] uses the Hartree-Fock-Bogoliubov plus Bardeen−Cooper−Schrieffer approach in a parity-independent way. The second model[62] was adopted by the RIPL3 library[63] (and is therefore labeled as such) and it uses a Hartree-Fock-Bogoliubov plus combinatorial approach. In Fig. 3a we show an optimized normalization of the RIPL3 model, which matches the experimental data. Similar discrepancies from the adopted normalization used in RIPL3 have been observed in other nuclei previously[64]. Since the experimental data do not extend all the

way to the neutron-separation energy ($S_n$=8.8 MeV), all three models shown in Fig. 3 were used to extract the $^{59}$Fe(n,γ)$^{60}$Fe reaction cross section and the results were included in the uncertainty band shown in Fig. 1a.

Fixing the NLD slope allowed for the gSF slope to also be fixed, since in the Oslo method the slopes of the two quantities are directly linked. The final step in our analysis was the absolute normalization of the gSF. This normalization was taken from Uberseder et al.[34] in the energy region between 5.4 and 6.5 MeV. The resulting gSF is shown in Fig. 1b of the main article.

The extracted NLD and gSF were used as input in the TALYS1.95 statistical model code[65] in order to extract the Maxwellian Averaged Cross Section (MACS). The MACS results are shown in Fig. 1a of the main article. The uncertainty band includes analysis and statistical uncertainties, as well as uncertainties from the NLD extrapolation mentioned above and the gSF normalization. The final MACS results for the relevant temperatures are shown in Table 1, while the final reaction rates are shown in Table 2.

## Data availability
The data sets generated during and/or analyzed during the current study are available from the corresponding author on request. Source data are provided with this paper in the Source Data file. The two-dimensional matrix produced and used in this work (Fig. 3a) has been deposited in the Zenodo database, under accession code https://doi.org/10.5281/zenodo.13785761. The recommended reaction rate can be found at https://doi.org/10.5281/zenodo.13799739. Source data are provided in this paper.

## Code availability
The codes used for the analysis of the present data are publicly available on github: https://github.com/oslocyclotronlab/oslo-method-software. The astrophysical calculations were performed with MESA which is an open-source code: https://docs.mesastar.org/en/24.08.1/. The theoretical calculations for the reaction rate extraction were performed with the open-source code TALYS: https://nds.iaea.org/talys/.

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

## Acknowledgements

The research was supported by Michigan State University, the National Superconducting Cyclotron Laboratory and the Facility for Rare Isotope Beams. The work was supported by the National Science Foundation under grants PHY 1913554, PHY 2209429 (A.Sp., H.S., A.P.K., K.H., A.D.), PHY 1565546 (A.Sp., S.N.L., H.S., C.S.), PHY-1848177 (CAREER) (A.Sp.), and PHY 1613188 (PDY). A.C., S.M., C. F., P.G. and C.P. were supported by the U.S. Department of Energy through the Los Alamos National Laboratory. Los Alamos National Laboratory is operated by Triad National Security, LLC, for the National Nuclear Security Administration of the U.S. Department of Energy (Contract No. 89233218CNA000001). The research presented in this article was supported by the Laboratory Directed Research and Development program of Los Alamos National Laboratory under project numbers 20160173ER and 20210808PRD1(C.F.). This material is based upon work supported in part by the U.S. Department of Energy, Office of Science, Office of Nuclear Physics under Contracts No. DE-SC0020451(S.N.L.) and DE-SC0023633 (S.N.L.), DE-SC0014285 (P.G.), and the National Nuclear Security Administration under Award No. DE-NA0003180 (B.P.C., K.C.) and the Stewardship Science Academic Alliances program through DOE Awards No DOE-DE-NA0003906 (D.R., R.L.). his work was performed under the auspices of the U.S. Department of Energy by Lawrence Livermore National Laboratory under Contract DE-AC52-07NA27344 (A.Sw.). A.C.L gratefully acknowledges funding by the European Research Council through ERC-STG-2014 under Grant Agreement No. 637686, from the Research Council of Norway grant number 316116, and the Norwegian Nuclear Center (project number 341985). A.C.L and J.E.M. gratefully acknowledge financial support by the Fulbright Program, which is sponsored by the U.S. Department of State and the U.S.-Norway Fulbright Foundation. S. L. was supported by the Laboratory Directed Research and Development Program at Pacific Northwest National Laboratory operated by Battelle for the U.S. Department of Energy.

## Author contributions

A. Spyrou prepared the proposal, ran the experiment, supervised the whole analysis and also performed parts of the analysis, led the interpretation, and prepared the manuscript. D.R. ran the experiment and performed the majority of the analysis. A.C. prepared the proposal and manuscript and participated in the experiment, analysis, and interpretation. C.F. and K.H. performed astrophysical calculations and contributed to the interpretation and manuscript preparation. S.N.L. prepared the proposal and manuscript and participated in the experiment, analysis, and interpretation. D.M. and H.S. contributed to the interpretation of the results. K.C., B.C., P.D., A.C.D., P.G., M.G., A.C.L., R.L., S.L., J.M., S.M., F.N., A.P.K., G.P., C.P., M.K.S., C.S., and A. Sweet participated in the experiment preparation and execution and contributed to the manuscript preparation.

## Competing interests

The authors declare no competing interests.
