## [Peer Review file · Nature Communications]

Enhanced production of ^{60}Fe in massive stars

Corresponding Author: Dr Artemis Spyrou

Version 0:

Reviewer comments:

Reviewer #1

(Remarks to the Author)

The present paper aims at resolving the long-standing discrepancy found in the literature between the observed $^{60}\text{Fe}/^{26}\text{Al}$ ratio and astrophysical model predictions, based on yields of massive star supernova ejecta. As they point out, a key issue is the uncertain $^{59}\text{Fe}(n,\gamma)$ reaction, and they do a marvelous job with their study based on improvements of the gamma-strength function determination at low energies with the aid of the beta-Oslo method. Its upbend at lowest energies results in an enhancement of the neutron-capture rate by about a factor of 1.5 over recent investigations (if interpreting their fig.1a correctly for the Yan et al. 2021 paper) and by about a factor of 2, when comparing to the Reaclib database utilized in most stellar evolution and explosion calculations. This is an excellent outcome determined with the most advanced methods in cross section determinations and should clearly be published in a highly prominent journal.

Nevertheless, I would recommend to re-investigate a bit the conclusions drawn from this result. Two new experiments have an effect: (1) the enhancement of the beta-decay rate of ^{59}Fe by a factor of 3.5 (Gao et al. 2021), leading to a reduction of the ^{60}Fe production by about 40%, i.e. close to a factor of 2, and (2) the enhancement of the neutron-capture rate on ^{59}Fe by about a factor of 2 in the present paper.

My question with respect to Fig.2: Does the calculation (showing about a factor of 2 enhancement over the MESA default) include the Reaclib (Rauscher and Thielemann 2000) neutron capture rate "plus" also already the enhanced ^{59}Fe beta decay or still the old ^{59}Fe beta-decay rate??

If not, both effects would cancel, leaving the factor of 3-10 overprediction of the $^{60}\text{Fe}/^{26}\text{Al}$ ratio in massive star models as cited on line 170 in the paper. This might then be on average still a factor of about 6. On the other hand, the authors cite for the observed $^{60}\text{Fe}/^{26}\text{Al}$ ratio the paper by Wang (2020) with 0.184 ± 0.042 , i.e. $0.142-0.226$. In the paper by Diehl et al. (2021) this is revised up to about 0.2-0.4, based apparently on taking into account in a better way the extra ^{60}Co radiation built up in INTEGRAL, due to long-term exposure to cosmic rays. This would reduce the discrepancy by about a factor 2. On the other hand, for $^{26}\text{Al}(n,p)$ and $^{26}\text{Al}(n,\alpha)$ there are indications that these cross sections might increase (Battino et al. 2023, MNRAS 520, 2436), which would reduce the ^{26}Al contribution, however with the caveat "Concerning explosive nucleosynthesis, an improvement of the current uncertainties between $T9 \sim 0.3$ and 2.5 is needed for future studies."

Reviewer #2

(Remarks to the Author)

The paper presents new measurement for the rate of the reaction $^{59}\text{Fe}(n,\gamma)^{60}\text{Fe}$. They obtain larger rates than previous values. Using their new rates for computing stellar models with MESA, they obtain larger abundance of ^{60}Fe in models of 15, 20 and 25 solar masses at solar metallicity.

They discuss a consequence of this larger ^{60}Fe production arguing that it will increase the discrepancy between observed and predicted $^{60}\text{Fe}/^{26}\text{Al}$ ratios, making thus the problem still more severe. They conclude that the problem cannot be solved by nuclear physics.

The paper is interesting but at least to my eyes remain too superficial for being published in Nature Astronomy.

The most solid parts of the paper are the new results for the neutron capture reaction rate of $^{59}\text{Fe}+n$, and the results of the stellar models when the new rates are used.

The weak parts are the discussion of the observations and the models required to make a theoretical estimate of it.

I agree with the authors that providing a more accurate rate for this reaction is a nice result, but their discussion about the consequences of this new rate on the predictions for the measured $^{60}\text{Fe}/^{26}\text{Al}$ in the interstellar medium deduced from gamma ray observations is too schematic. It is based only of a few models but not on a model trying to reproduce the observed ratio.

So I cannot recommend the scientific editor to accept this manuscript for publication in Nature Communications.

I give below a few specific points that leads me to that conclusion (they are not at the same level of importance and are just ordered here as they appeared reading the paper sequentially). On the whole they indicate why I think, in its present state, is not appropriate for publication.

1) First sentence of the introduction

Unique signatures of supernova explosions can come from long-lived radioisotopes.
These freshly synthesized isotopes...

Depending on how we understand "long-lived" in the first sentence, it appears a bit inconsistent with the beginning of the next sentence "These freshly...", indeed if they have a long lifetime the radioisotopes may have been ejected a long time ago and thus they were not freshly synthesized.

May be, it would be better to say radioisotope with a lifetime between 105 and 106 years?

2) Second sentence of introduction

...or emit radiation that can be detected inside the solar system

This is correct in the sense that every observation is made from inside the solar system but the sentence may be misleading in the sense that it may lead to be interpreted as if sources of emissions would be located in the solar system. The gamma ray emission from decay of iron 60 comes from the iron 60 of the interstellar medium of the Milky Way most of it being located in the central/disk regions of our Galaxy. Same for ^{26}Al .

3) Introduction (line 44)

They have been detected in a number of Earth and space-based measurements: stardust, cosmic 45 rays, γ -ray measurements of radioactivities in the galaxy, material deposited in deep-ocean crusts and on the surface of the moon, indirect signatures in meteorites of their presence during the formation of the solar system, and more...

May be it would be better to replace stardust by meteoritic pre-solar grains.

4) Introduction line 48

$^{60}\text{Fe}/^{26}\text{Al}$ ratio has been identified as an excellent probe to study nucleosynthesis in massive
49 stars

This assessment is too general. It would be fair to be more precise here and to indicate which measures of this ratio can shed light on the nucleosynthesis in massive stars. Are the authors considering the measured ratio in the interstellar medium coming from the detections of the gamma-rays emitted by the decay of these two radioisotopes? Are they considering the ratio obtained from inclusions in meteorites, or are they considering this ratio in pre-solar grains? Each of these measurements involve differently stellar nucleosynthesis and thus provide different types of constraints. It would be nice for the reader to know which topic will be addressed here.

5) Introduction line 54

While accurate measurements of this ratio exist in the Galaxy and the early solar
system⁹, models tend to strongly overestimate it^{22,24}

Reading the abstract of the reference 24, indicated by the authors: see except of the abstract of 24 here:

...By means of these yields we try to reproduce two quite strong observational constraints related to the abundances of these nuclei in the interstellar medium, i.e., the number of 1.8 photons per Lyman continuum photon, RGxL , and the $^{60}\text{Fe}/^{26}\text{Al}$ -ray line flux ratio. RGxL is found to be roughly constant along the Galactic plane (and of the order of 1:25 ; 1011), while the $^{60}\text{Fe}/^{26}\text{Al}$ ratio has been recently measured by both RHESSI (0:17 0:05) and SPI (INTEGRAL) (0:11 0:03). We can quite successfully fit simultaneously both ratios for a quite large range of exponents of the power-law initial mass function....

So I am not sure I understand what the authors mean by strongly overestimate it.

6) Introduction, line 62

H-burning material can be ejected earlier due to stellar winds, which adds to the information provided by the ^{26}Al detection.

What do the authors mean by ... which adds to the information provided by the ^{26}Al detection.?

7) Introduction, line 66

...giving direct access to the stellar yield ratio right before the supernova explosion

Not sure I understand this sentence: why right before?

Discussion of $^{59}\text{Fe}(n, \gamma)^{60}\text{Fe}$ reaction

8) line 85

It consists of a series of neutron-capture reactions starting at the stable isotope ^{58}Fe , and depending on the astrophysical conditions a competition between β decays and or (p,n) reactions

The sentence does not appear to me well constructed. Do the authors want to say

The outcome of the nuclear network depends on the astrophysical conditions especially on the competition between β decays and or (p,n) reactions?

9) line 104

with the extrapolation of the previous ^{60}Fe study³⁴

Is it the continuous black line? I guess yes looking at the reference.

10) line 118

For comparison purposes, we used the lower limit of our results

I do not understand for what they used the lower limit?

and removed the low-energy enhancement from the MACS calculation (Fig. 1a, purple solid line).

I do not understand this sentence and the meaning of the purple line that is in the Uberseder 2014 range.

This part would deserve to be more clearly written. The caption of the figure is fine, but the text could be made clearer.

Implications on $^{60}\text{Fe}/^{26}\text{Al}$ ratio in the galaxy

This paragraph appears to me as being the weakest in the paper.

11) line 175

...pointing to deeper issues in our understanding of supernovae.

This may indeed be the case that our understanding of supernovae is not enough robust for allowing reliable predictions. But the ratio of $^{60}\text{Fe}/^{26}\text{Al}$ measured in the interstellar medium from gamma-ray line measurements, do not only depend on the physics of supernovae. It also depend on various other inputs as for instance, the variation of the metallicity and of the star formation rate with the galactocentric distance, the physics of wind ejection (for ^{26}Al at least), may be the possible impact of close binaries... it involves computing the contribution of stars of different initial masses born in the last few million years in the galaxy (see for instance Wang et al. 2020, *The Astrophysical Journal*, Volume 889, Issue 2, id.169).

Thus, while the statement above is not wrong it is in my view incomplete. Actually, the authors are aware of at least some of these aspects as testify by the paragraph that follows. But in a more concrete way their conclusion is not based on quantitative assessments of these different aspects. In brief to really conclude that the physics of supernovae is the only cause, then other sources of uncertainties should be eliminate. This is not really done here except on the possible uncertainty linked to the $^{59}\text{Fe} + n$ reaction.

12) line 186

Therefore, the increased production of ^{60}Fe found in the present work may be explained by a higher mass limit for supernova explosions.

I do not understand this sentence. The paper shows that the increased production is due to the use of the new rate of the neutron capture reaction on ^{59}Fe not to a higher mass limit.

May be the authors want to write that an increased production ... may also be due to a higher upper mass limit for supernova explosion?

Or that the increased production when using the new rates may be mitigated by considering lower values for the upper mass limit of stars exploding as supernovae?

13) line 195

Therefore, the increased ^{60}Fe production found in the present work, which causes an even higher $^{60}\text{Fe}/^{26}\text{Al}$ ratio than previously predicted, implies that a wide-scale adoption of rotating star models might not be appropriate until the $^{60}\text{Fe}/^{26}\text{Al}$ ratio puzzle is solved.

This is my view a not very strong conclusion. As mentioned above, to reproduce the observed $^{60}\text{Fe}/^{26}\text{Al}$ requires many more elements than just choosing between non-rotating and rotating models starting with a given initial rotation. It is not possible in my view to support such a conclusion without a more detailed study looking at all other possibilities.

Summary and conclusion

210 The solution to the puzzle must come from the stellar modeling by, for example, reducing stellar rotation or by increasing the explosibility limit for massive stars.

If still more massive stars explode as supernovae will it not make the problem worse, by adding ^{60}Fe ? While I agree that stellar models are still in need of being improved and that knowing better one important reaction rate is a step towards this aim, the discussion here is very schematic and lacks some more detailed and quantitative arguments based on exploring the impacts of different aspects of the problem.

Reviewer #3

(Remarks to the Author)

The manuscript "Enhanced production of ^{60}Fe in massive stars" by A. Spyrou and collaborators describes the implications of recent results from nuclear experiments that suggest an enhancement of the gamma-ray strength function at low energies for ^{60}Fe . This enhances the main production cross section for ^{60}Fe in massive stars, exacerbating the existing tension between theoretical models and observations of the $^{26}\text{Al}/^{60}\text{Fe}$ ratio based on gamma-ray observations in the galaxy.

The authors demonstrate the effect of the change of the nuclear reaction rate with stellar evolution calculations and discuss the impact of rotation on the production of ^{60}Fe and ^{26}Al in stars. They conclude that nuclear physics uncertainties cannot provide a solution to the discrepancy between $^{26}\text{Al}/^{60}\text{Fe}$ observations and theoretical models.

The article is well written and concise. The presented results are original and noteworthy. The presentation of the results is adequate for a wider scientific audience. The demonstration of the impact with stellar evolution calculations is very convincing. The experiment and analysis methods are appropriately described in the supplemental material. The results of the manuscript are of broad and general interest for Nuclear Physics, Astrophysics and Cosmochemistry. The work clearly shows the need for revisions and reconsiderations in stellar evolution models, which is relevant for a wide range of researchers. Firm evidence for the existence of low-energy enhancement of GSFs is also very relevant for the wider nuclear physics community. I recommend the publication of the manuscript.

Below are minor recommendations and questions

-- Line 66: "before the SN explosion": Both ^{26}Al and ^{60}Fe are affected by the supernova explosion and since they are produced in different regions of the star they are affected in different ways. Jones et al. 2019 (MNRAS, 485, p.4287) have demonstrated that ^{60}Fe changes by up to a factor 2 depending on the explosion energy. While this uncertainty cannot be addressed in the current manuscript, I think it would be appropriate to mention this aspect.

-- Line 208: The authors claim "With our new result, the nuclear physics uncertainties for the production of both isotopes are well under control". However, there is a large sensitivity of stellar evolution to other nuclear reactions that also affects ^{60}Fe . For example Tur et al. 2010 (The Astrophysical Journal, Volume 718, Issue 1, pp. 357) have shown that the ^{60}Fe yields of massive stars can change by more than a factor two when the $^3\alpha$ and $^{12}\text{C}(\alpha, n)$ reaction rates are changed within the uncertainties. While this does not diminish the importance of the presented work at all, I recommend rephrasing the sentence slightly: There are still nuclear physics uncertainties that can affect ^{60}Fe and ^{26}Al .

-- If the upbend of the GSF is a general phenomenon, is it possible or likely that an upbend also exists for ^{60}Fe . If so, an increase in the $^{60}\text{Fe}(n, g)$ reaction rate could compensate for the reported increase in the $^{59}\text{Fe}(n, g)$ rate. Is this a possibility?

Version 1:

Reviewer comments:

Reviewer #1

(Remarks to the Author)

The authors have addressed the points I made in my first report. Please accept this revised version.

Reviewer #2

(Remarks to the Author)

The authors have made changes that in my view improves the paper. I would therefore recommend the scientific editor to accept to publish this paper in this present version. My main justification is that the paper presents indeed improved determination of a key nuclear reaction rate. My main critics were mainly driven by some parts of the text describing more astrophysical aspects of the whole problem. But now these aspects have been included in this new revised version of the paper.

Reviewer #3

(Remarks to the Author)

I am satisfied with the author's responses and changes the manuscript.

We would like to thank all three referees for their careful review of the manuscript and their useful comments. We have addressed all of them below and feel that the new version of the manuscript is improved because of them.

REVIEWER COMMENTS

Reviewer #1 (Remarks to the Author):

The present paper aims at resolving the long-standing discrepancy found in the literature between the observed $^{60}\text{Fe}/^{26}\text{Al}$ ratio and astrophysical model predictions, based on yields of massive star supernova ejecta. As they point out, a key issue is the uncertain $^{59}\text{Fe}(n,\gamma)$ reaction, and they do a marvelous job with their study based on improvements of the gamma-strength function determination at low energies with the aid of the beta-Oslo method. Its upbend at lowest energies results in an enhancement of the neutron-capture rate by about a factor of 1.5 over recent investigations (if interpreting their fig.1a correctly for the Yan et al. 2021 paper) and by about a factor of 2, when comparing to the Reaclib database utilized in most stellar evolution and explosion calculations. This is an excellent outcome determined with the most advanced methods in cross section determinations and should clearly be published in a highly prominent journal.

Nevertheless, I would recommend to re-investigate a bit the conclusions drawn from this result. Two new experiments have an effect: (1) the enhancement of the beta-decay rate of ^{59}Fe by a factor of 3.5 (Gao et al. 2021), leading to a reduction of the ^{60}Fe production by about 40%, i.e. close to a factor of 2, and (2) the enhancement of the neutron-capture rate on ^{59}Fe by about a factor of 2 in the present paper.

My question with respect to Fig.2: Does the calculation (showing about a factor of 2 enhancement over the MESA default) include the Reaclib (Rauscher and Thielemann 2000) neutron capture rate “plus” also already the enhanced ^{59}Fe beta decay or still the old ^{59}Fe beta-decay rate??

If not, both effects would cancel, leaving the factor of 3-10 overprediction of the $^{60}\text{Fe}/^{26}\text{Al}$ ratio in massive star models as cited on line 170 in the paper. This might then be on average still a factor of about 6. On the other hand, the authors cite for the observed $^{60}\text{Fe}/^{26}\text{Al}$ ratio the paper by Wang (2020) with 0.184 ± 0.042 , i.e. 0.142-0.226. In the paper by Diehl et al. (2021) this is revised up to about 0.2-0.4, based apparently on taking into account in a better way the extra ^{60}Co radiation built up in INTEGRAL, due to long-term exposure to cosmic rays. This would reduce the discrepancy by about a factor 2.

On the other hand, for $^{26}\text{Al}(n,p)$ and $^{26}\text{Al}(n,\alpha)$ there are indications that these cross sections might increase (Battino et al. 2023, MNRAS 520, 2436), which would reduce the ^{26}Al contribution, however with the caveat "Concerning explosive nucleosynthesis, an improvement of the current uncertainties between T9~0.3 and 2.5 is needed for future studies."

We completely agree with the referee that the picture is more complex and there are more parameters affecting the $^{60}\text{Fe}/^{26}\text{Al}$ ratio. This is the reason we did not attempt to extract an absolute value for the ratio, but rather focused on investigating the impact of the new reaction rate on the ^{60}Fe production compared to the previous standard. In that respect, the use or not of the new beta-decay rate will not change the conclusions of the present work since the increased

(n,g) reaction rate still produces a factor of 2 more ^{60}Fe than the previous value of this rate. We do mention in the manuscript that the new beta-decay rate by Gao et al. brings the ^{60}Fe production lower, and hence the $^{60}\text{Fe}/^{26}\text{Al}$ ratio prediction closer to the observed value. However, our result has the opposite effect, hence we are back to the large discrepancy (if not worse). The new beta-decay rate was not part of our calculations. We used MESA, which uses REACLIB for the reaction rates, and only changed the $^{59}\text{Fe}(n,g)^{60}\text{Fe}$ reaction rate. Once published we will recommend the new rate (and the beta-decay rate) to be adopted by REACLIB so that they can be more broadly used in astrophysical calculations.

We also agree with the referee that there still remain other uncertainties in the nuclear reactions and for this reason we modified the relevant sentence (Line 216):

“While uncertainties in the nuclear physics aspects still remain, our result removes one of the most significant uncertainties in the ^{60}Fe production.”

Reviewer #2 (Remarks to the Author):

The paper presents new measurement for the rate of the reaction $^{59}\text{Fe}(n, \gamma)^{60}\text{Fe}$. They obtain larger rates than previous values. Using their new rates for computing stellar models with MESA, they obtain larger abundance of ^{60}Fe in models of 15, 20 and 25 solar masses at solar metallicity.

They discuss a consequence of this larger ^{60}Fe production arguing that it will increase the discrepancy between observed and predicted $^{60}\text{Fe}/^{26}\text{Al}$ ratios, making thus the problem still more severe. They conclude that the problem cannot be solved by nuclear physics.

The paper is interesting but at least to my eyes remain too superficial for being published in Nature Astronomy.

The most solid parts of the paper are the new results for the neutron capture reaction rate of $^{59}\text{Fe}+n$, and the results of the stellar models when the new rates are used.

The weak parts are the discussion of the observations and the models required to make a theoretical estimate of it.

I agree with the authors that providing a more accurate rate for this reaction is a nice result, but their discussion about the consequences of this new rate on the predictions for the measured $^{60}\text{Fe}/^{26}\text{Al}$ in the interstellar medium deduced from gamma ray observations is too schematic. It is based only of a few models but not on a model trying to reproduce the observed ratio.

So I cannot recommend the scientific editor to accept this manuscript for publication in Nature Communications.

I give below a few specific points that leads me to that conclusion (they are not at the same level of importance and are just ordered here as they appeared reading the paper sequentially). On the whole they indicate why I think, in its present state, is not appropriate for publication.

1) First sentence of the introduction

Unique signatures of supernova explosions can come from long-lived radioisotopes.
These freshly synthesized isotopes...

Depending on how we understand "long-lived" in the first sentence, it appears a bit inconsistent with the beginning of the next sentence "These freshly...", Indeed if they have a long lifetime the radioisotopes may have been ejected a long time ago and thus they were not freshly synthesized.

May be, it would be better to say radioisotope with a lifetime between 10^5 and 10^6 years?

We have changed the sentence to: "Unique signatures of supernova explosions can come from long-lived radioisotopes (with half-lives of the order of a million years)." We have also added quotes to the word "freshly" to indicate that this refers to astrophysical time scales.

2) Second sentence of introduction

...or emit radiation that can be detected inside the solar system

This is correct in the sense that every observation is made from inside the solar system but the sentence may be misleading in the sense that it may lead to be interpreted as if sources of emissions would be located in the solar system. The gamma ray emission from decay of iron 60 comes from the iron 60 of the interstellar medium of the Milky Way most of it being located in the central/disk regions of our Galaxy. Same for ^{26}Al .

We agree that this part of the sentence can be misinterpreted. We have modified the text to hopefully make it more clear:

"...to either travel all the way to the solar system or emit radiation that does".

3) Introduction (line 44)

They have been detected in a number of Earth and space-based measurements: stardust, cosmic rays, γ -ray measurements of radioactivities in the galaxy, material deposited in deep-ocean crusts and on the surface of the moon, indirect signatures in meteorites of their presence during the formation of the solar system, and more...

May be it would be better to replace stardust by meteoritic pre-solar grains.

Both terms have been used in the literature. Looking back to the language used in the relevant reference (Groopman et al), they use “presolar stardust grains” so we adopted this term in the manuscript for consistency.

4) Introduction line 48

$^{60}\text{Fe}/^{26}\text{Al}$ ratio has been identified as an excellent probe to study nucleosynthesis in massive stars

This assessment is too general. It would be fair to be more precise here and to indicate which measures of this ratio can shed light on the nucleosynthesis in massive stars. Are the authors considering the measured ratio in the interstellar medium coming from the detections of the gamma-rays emitted by the decay of these two radioisotopes? Are they considering the ratio obtained from inclusions in meteorites, or are they considering this ratio in pre-solar grains? Each of these measurements involve differently stellar nucleosynthesis and thus provide different types of constraints. It would be nice for the reader to know which topic will be addressed here.

We agree with the referee that this is a general statement, and it was meant as such since this is the introductory sentence to the topic of the $^{60}\text{Fe}/^{26}\text{Al}$ ratio. Based on the referee’s suggestion we have included the relevant clarification in the next sentence: “This ratio, extracted from γ -ray observations, can constrain...”

5) Introduction line 54

While accurate measurements of this ratio exist in the Galaxy and the early solar system, models tend to strongly overestimate it

Reading the abstract of the reference 24, indicated by the authors: see except of the abstract of 24 here:

...By means of these yields we try to reproduce two quite strong observational constraints related to the abundances of these nuclei in the interstellar medium, i.e., the number of 1.8 photons per Lyman continuum photon, RGxL , and the $^{60}\text{Fe}/^{26}\text{Al}$ -ray line flux ratio. RGxL is found to be roughly constant along the Galactic plane (and of the order of 1:25 ; 1011), while the $^{60}\text{Fe}/^{26}\text{Al}$ ratio has been recently measured by both RHESSI (0:17 0:05) and SPI (INTEGRAL) (0:11 0:03). We can quite successfully fit simultaneously both ratios for a quite large range of exponents of the power-law initial mass function....

So I am not sure I understand what the authors mean by strongly overestimate it.

We agree with the referee that the abstract of Ref. 24 indicates that under certain conditions and assumptions the particular model could reproduce the $^{60}\text{Fe}/^{26}\text{Al}$ ratio. However, in their paper the authors present different quantities that can affect this ratio, such as IMF slope and M_{top} (explodability). Most of their calculations overpredict the ratio, although indeed some of them are in agreement. This is the only study where this is the case. All other calculations overpredict

the ratio. We therefore find that the general introductory statement mentioned above is still overall true.

6) Introduction, line 62

H-burning material can be ejected earlier due to stellar winds, which adds to the information provided by the ^{26}Al detection.

What do the authors mean by ... which adds to the information provided by the ^{26}Al detection.?

With this sentence we mean that because ^{26}Al is also ejected during stellar winds, when ^{26}Al is detected we can also learn something about that aspect of stellar evolution. However, we can see the referee's point that this sentence is unclear. The sentence was rephrased to make it more simple: "H-burning material can be ejected earlier due to stellar winds, which adds to the ^{26}Al yield".

7) Introduction, line 66

...giving direct access to the stellar yield ratio right before the supernova explosion

Not sure I understand this sentence: why right before?

Once again the referee has a valid point about the clarity of the used language. The sentence was meant to point out that the measured ratio will not be affected by other external parameters related to the source (like its distance) and will therefore be representative of the production of the two isotopes by that source. It shouldn't say "right before" the supernova, but "right after" the supernova, to include the production before and during the supernova. This sentence was changed in the manuscript.

Discussion of $^{59}\text{Fe}(n, \gamma)^{60}\text{Fe}$ reaction

8) line 85

It consists of a series of neutron-capture reactions starting at the stable isotope ^{58}Fe , and depending on the astrophysical conditions a competition between β decays and or (p,n) reactions

The sentence does not appear to me well constructed. Do the authors want to say

The outcome of the nuclear network depends on the astrophysical conditions especially on the competition between β decays and or (p,n) reactions?

This sentence refers to the competition between neutron capture and β -decay OR between neutron capture and (p,n) reactions. One or the other will dominate depending on the astrophysical conditions. This was shown in Figures 29 and 30 of Diehl et al. 2021. We have rephrased the sentence to make it more clear:

“It consists of a series of neutron-capture reactions starting at the stable isotope ^{58}Fe , which compete with either β decays or (p,n) reactions, depending on the astrophysical conditions³¹.”

9) line 104

with the extrapolation of the previous ^{60}Fe study

Is it the continuous black line? I guess yes looking at the reference.

We have modified the text to make it more clear: “with the extrapolation of the previous ^{60}Fe study (solid black line)...”

10) line 118

For comparison purposes, we used the lower limit of our results

I do not understand for what they used the lower limit?

and removed the low-energy enhancement from the MACS calculation (Fig. 1a, purple solid line).

I do not understand this sentence and the meaning of the purple line that is in the Uberseder 2014 range.

This part would deserve to be more clearly written. The caption of the figure is fine, but the text could be made clearer.

The purpose of the purple line was to show that indeed, the main difference between the Uberseder result and the present work is the presence of the upbend. Using the Uberseder gSF (no upbend) in our calculations results in a MACS that is in agreement with the Uberseder result. This was just a sanity check to show that no other discrepancies between the two measurements exist.

Based on the referee’s comments it is clear that we did not do a good job explaining this. The text was modified as follows:

“As a consistency check, we performed the MACS calculation removing the low-energy enhancement from our gSF (Fig. 1a, purple solid line) and keeping all other parameters the same. This calculation enables a more comparison with the Uberseder *et al.* measurement. Although other parameters, like the NLD, are not identical, it can be seen in Fig. 1a that by removing the low-energy enhancement the two measurements are consistent. The main difference between our result and the one from Uberseder *et al.* comes from the fact that the latter measurement was not sensitive to the presence of the low-energy enhancement.”

Implications on $^{60}\text{Fe}/^{26}\text{Al}$ ratio in the galaxy

This paragraph appears to me as being the weakest in the paper.

11) line 175

...pointing to deeper issues in our understanding of supernovae.

This may indeed be the case that our understanding of supernovae is not enough robust for allowing reliable predictions. But the ratio of $^{60}\text{Fe}/^{26}\text{Al}$ measured in the interstellar medium from gamma-ray line measurements, do not only depend on the physics of supernovae. It also depend on various other inputs as for instance, the variation of the metallicity and of the star formation rate with the galactocentric distance, the physics of wind ejection (for ^{26}Al at least), may be the possible impact of close binaries... it involves computing the contribution of stars of different initial masses born in the last few million years in the galaxy (see for instance Wang et al. 2020, The Astrophysical Journal, Volume 889, Issue 2, id.169).

Thus, while the statement above is not wrong it is in my view incomplete. Actually, the authors are aware of at least some of these aspects as testify by the paragraph that follows. But in a more concrete way their conclusion is not based on quantitative assessments of these different aspects. In brief to really conclude that the physics of supernovae is the only cause, then other sources of uncertainties should be eliminate. This is not really done here except on the possible uncertainty linked to the $^{59}\text{Fe} + n$ reaction.

The referee is of course correct that the statement above is oversimplified. The main point of the sentence was to say that the discrepancy between models and observations in the $^{60}\text{Fe}/^{26}\text{Al}$ ratio will probably not be resolved through nuclear physics. Pointing only to supernova issues is indeed not accurate. We have modified the single sentence to two sentences as follows: "The firmer constraints on the nuclear reaction network from the present work show that a nuclear solution to this problem is unlikely. This suggests that the solution to the puzzle should come from the description of the astrophysical environment and processes that affect the two isotopes."

12) line 186

Therefore, the increased production of ^{60}Fe found in the present work may be explained by a higher mass limit for supernova explosions.

I do not understand this sentence. The paper shows that the increased production is due to the use of the new rate of the neutron capture reaction on ^{59}Fe not to a higher mass limit.

May be the authors want to write that an increased production ... may also be due to a higher upper mass limit for supernova explosion?

Or that the increased production when using the new rates may be mitigated by considering lower values for the upper mass limit of stars exploding as supernovae?

We have modified the above sentence as follows to make the meaning more clear: "Therefore, the increased production of ^{60}Fe found in the present work can be balanced by assuming a lower mass limit for supernova explosions in the models."

13) line 195

Therefore, the increased ^{60}Fe production found in the present work, which causes an even higher $^{60}\text{Fe}/^{26}\text{Al}$ ratio than previously predicted, implies that a wide-scale adoption of rotating star models might not be appropriate until the $^{60}\text{Fe}/^{26}\text{Al}$ ratio puzzle is solved.

This is my view a not very strong conclusion. As mentioned above, to reproduce the observed $^{60}\text{Fe}/^{26}\text{Al}$ requires many more elements than just choosing between non-rotating and rotating models starting with a given initial rotation. It is not possible in my view to support such a conclusion without a more detailed study looking at all other possibilities.

We agree with the referee that a strong conclusion for choosing between rotating and non-rotating models cannot be drawn here. However, since the $^{60}\text{Fe}/^{26}\text{Al}$ ratio was indeed shown to be impacted by rotation, the connection we try to make here is valid. We have rephrased the sentence to make it less strong:

“Therefore, the increased ^{60}Fe production found in the present work, which causes an even higher $^{60}\text{Fe}/^{26}\text{Al}$ ratio than previously predicted, cannot be explained by introducing stellar rotation to the models.”

Summary and conclusion

The solution to the puzzle must come from the stellar modeling by, for example, reducing stellar rotation or by increasing the explodability limit for massive stars.

If still more massive stars explode as supernovae will it not make the problem worse, by adding ^{60}Fe ? While I agree that stellar models are still in needs of being improved and that knowing better one important reaction rate is a step towards this aim, the discussion here is very schematic and lacks some more detailed and quantitative arguments based on exploring the impacts of different aspects of the problem.

First of all, the referee is absolutely right. We accidentally flipped the dependency of the $^{60}\text{Fe}/^{26}\text{Al}$ ratio on the explodability limit. This was corrected in comment 12 and here. We also agree with the referee that the discussion on the $^{60}\text{Fe}/^{26}\text{Al}$ ratio could be expanded through additional calculations and investigations. However, doing so would introduce a model dependency that we wanted to avoid in the present work. All conclusions are based on the new measurement, the impact of the measurement on ^{60}Fe production and on literature discussions of what a higher ^{60}Fe production might mean. We did not find in the literature additional investigations that can affect the $^{60}\text{Fe}/^{26}\text{Al}$ ratio directly. We hope that our result will inspire new studies within the massive star community to resolve the discrepancy. Therefore, we agree that the discussion is more qualitative, however we believe that a more quantitative analysis using different massive star and supernova models and different parameters is beyond the scope of this work.

The concluding sentence above was corrected to:

“The solution to the puzzle must come from the stellar modeling by, for example, reducing stellar rotation, assuming smaller explosibility mass limits for massive stars, or modifying other stellar parameters.”

Reviewer #3 (Remarks to the Author):

The manuscript "Enhanced production of ^{60}Fe in massive stars" by A. Spyrou and collaborators describes the implications of recent results from nuclear experiments that suggest an enhancement of the gamma-ray strength function at low energies for ^{60}Fe . This enhances of the main production cross section for ^{60}Fe in massive stars, exacerbating the existing tension between theoretical models and observations of the $^{26}\text{Al}/^{60}\text{Fe}$ ratio based on gamma-ray observations in the galaxy.

The authors demonstrate the effect of the change of the nuclear reaction rate with stellar evolution calculations and discuss the impact of rotation on the production of ^{60}Fe and ^{26}Al in stars. They conclude that nuclear physics uncertainties cannot provide a solution to the discrepancy between $^{26}\text{Al}/^{60}\text{Fe}$ observations and theoretical models.

The article is well written and concise. The presented results are original and noteworthy. The presentation of the results is adequate for a wider scientific audience. The demonstration of the impact with stellar evolution calculations is very convincing.

The experiment and analysis methods are appropriately described in the supplemental material. The results of the manuscript are of broad and general interest for Nuclear Physics, Astrophysics and Cosmochemistry. The work clearly shows the need for revisions and reconsiderations in stellar evolution models, which is relevant for a wide range of researchers. Firm evidence for the existence of low-energy enhancement of GSFs is also very relevant for the wider nuclear physics community.

I recommend the publication of the manuscript.

Below are minor recommendations and questions

-- Line 66: "before the SN explosion": Both ^{26}Al and ^{60}Fe are affected by the supernova explosion and since they are produced in different regions of the star they are affected in different ways. Jones et al. 2019 (MNRAS, 485, p.4287) have demonstrated that ^{60}Fe changes by up to a factor 2 depending on the explosion energy. While this uncertainty cannot be addressed in the current manuscript, I think it would be appropriate to mention this aspect.

In the manuscript we have a discussion of the different mechanisms and areas of the star that produce the two isotopes later in the manuscript, but the referee is correct that the explosion energy was not mentioned. This was because there is no clear conclusion we could draw. Figure 17 of Jones et al 2019 shows that for 15 solar-mass stars the $^{60}\text{Fe}/^{26}\text{Al}$ ratio increases for higher explosion energies, while for 20 solar-mass stars it decreases, and for 25 solar-mass stars it remains relatively flat. Nevertheless, since explosion energy does impact the $^{60}\text{Fe}/^{26}\text{Al}$ ratio, it should be mentioned and we have now added the following text:

“Additional stellar parameters can affect the $^{60}\text{Fe}/^{26}\text{Al}$ ratio and further investigations are needed to shed light on this puzzle. One example that was shown to have a significant impact is the explosion energy⁴⁰. Jones *et al.*⁴⁰ showed that the dependence of the $^{60}\text{Fe}/^{26}\text{Al}$ is different for 15, 20 or $25M_{\odot}$ stars, therefore additional studies are needed to understand the impact in the overall $^{60}\text{Fe}/^{26}\text{Al}$ ratio.”

-- Line 208: The authors claim "With our new result, the nuclear physics uncertainties for the production of both isotopes are well under control". However, there is a large sensitivity of stellar evolution to other nuclear reactions that also affects ^{60}Fe . For example Tur et al. 2010 (The Astrophysical Journal, Volume 718, Issue 1, pp. 357) have shown that the ^{60}Fe yields of massive stars can change by more than a factor two when the 3α and $^{12}\text{C}(\alpha, \text{g})$ reaction rates are changed within the uncertainties. While this does not diminish the importance of the presented work at all, I recommend rephrasing the sentence slightly: There are still nuclear physics uncertainties that can affect ^{60}Fe and ^{26}Al .

We agree with the referee that the sentence was oversimplifying the current status of the nuclear physics. We have rephrased the sentence to make it more fair:

“While uncertainties in the nuclear physics aspects still remain, our result removes one of the most significant uncertainties in the ^{60}Fe production.”

-- If the upbend of the GSF is a general phenomenon, is it possible or likely that an upbend also exists for ^{60}Fe . If so, an increase in the $^{60}\text{Fe}(\text{n}, \text{g})$ reaction rate could compensate for the reported increase in the $^{59}\text{Fe}(\text{n}, \text{g})$ rate. Is this a possibility?

The referee makes a valid argument that the upbend should most likely exist also in ^{61}Fe and would affect the cross section of the $^{60}\text{Fe}(\text{n}, \text{g})^{61}\text{Fe}$ reaction if estimated in an indirect way. However, this reaction cross section has been measured directly (Uberseder, et al, PRL 102 (2009) 15101) using the activation technique, therefore the existence or not of an upbend will not affect the results of that measurement.